# The Efficacy of Cervical Lordosis Rehabilitation for Nerve Root Function and Pain in Cervical Spondylotic Radiculopathy: A Randomized Trial with 2-Year Follow-Up

**DOI:** 10.3390/jcm11216515

**Published:** 2022-11-02

**Authors:** Ibrahim M. Moustafa, Aliaa A. Diab, Deed E. Harrison

**Affiliations:** 1Department of Physiotherapy, College of Health Sciences, University of Sharjah, Sharjah P.O. Box 27272, United Arab Emirates; 2Basic Science Department, Faculty of Physical Therapy, Cairo University, Giza 12613, Egypt; 3CBP Nonprofit—A Spine Research Foundation, Eagle, ID 83616, USA

**Keywords:** cervical spine, dermatomal somatosensory evoked potential, lordosis, randomized trial, spondylotic radiculopathy, traction

## Abstract

Sagittal cervical alignment is a clinically related feature in patients suffering from chronic cervical spondylotic radiculopathy (CSR). We designed this randomized trial to explore the effects of cervical lordosis (CL) correction in thirty chronic lower CSR patients with CL < 20°. Patients were assigned randomly into two equal groups, study (SG) and control (CG). Both groups received neck stretching and exercises and infrared radiation; additionally, the SG received cervical extension traction. Treatments were applied 3× per week for 10 weeks after which groups were followed for 3 months and 2 years. The amplitude of dermatomal somatosensory evoked potentials (DSSEPS), CL C2–C7, and pain scales (NRS) were measured. The SG had an increase in CL post-treatment (*p* < 0.0001), this was maintained at 3 months and 2 years. No statistical improvement in CL was found for the CG. A significant reduction in NRS for SG after 10 weeks of treatment with non-significant loss of change at 3 months and continued improvement at 2 years was found. CG had less significant improvement in post-treatment NRS; the 3-month and 2-year measures revealed significant worsening in NRS. An inverse linear correlation between increased CL and NRS was found (r = −0.49; *p* = 0.005) for both groups initially and maintained in SG at the final 2-year follow-up (r = −0.6; *p* = 0.01). At 10 weeks, we found significant improvements in DSSEPS for both groups (*p* < 0.0001). We identified a linear correlation between initial DSSEPs and CL for both groups (*p* < 0.0001), maintained only in the SG at the final follow-up for all levels (*p* < 0.0001). Improved CL in the SG correlated with significant improvements in nerve root function and pain rating in patients with CSR at short and long-term follow-up. These observed effects indicate that clinicians involved in the treatment of patients with symptoms of cervical degenerative disorders should add sagittal curve correction to their armamentarium of rehabilitation procedures for relevant patient populations.

## 1. Introduction

Cervical spondylotic radiculopathy (CSR) is one of the most common causes of cervical radiculopathy [1,2]. It has been documented that the incidence of cervical degenerative abnormalities increases with age having the greatest frequency in the fifth to sixth decade of life [3]. Spondylotic degenerative findings appear to be the most common followed by disc damage and these are most common in the lower cervical spine discs (C5–C6) [3,4]. The degenerative state of the intervertebral disc, vertebral body and adjacent structures, occurs as a result of several factors including segmental injury/trauma and alterations in the sagittal alignment of the cervical spine; including reductions in the segmental and total angle of cervical curvature [4,5,6,7].

Causation of CSR involves multiple factors but the mechanical compression and shear loads acting on the nerve roots result in inflammation and this is the primary driver of the pain, decreased cervical movement, and consequent neurological disturbances. CSR incidence increases with age and has an estimated frequency of 0.35% in the fifth to sixth decade of life [2]. Recently, multiple systematic literature reviews have been published seeking to understand the complexities of CSM and its natural history, conservative management strategies, and the need for surgical interventions [2,8,9,10,11,12,13]. For patients with intractable pain and with motor loss of less than three out of five, surgical intervention is warranted [2]. However, in comparison between conservative and surgical management trials, the long-term outcomes at 1–2 years generally show conservative care to be equally effective for less severe CSM patients [2,8,9,10,11,12].

Although there is general agreement regarding the need for conservative treatment for CSR, the precise treatment protocols for the best results and when to use them for CSR disorders still remain an enigma [2,8,9,10,11,12,13]. Conservative treatments for CSM include rehabilitative exercise therapy, mechanical cervical traction, transcutaneous electrical nerve stimulation, pain management, education, cervical collars, and spinal manipulative therapy [2,8,9,10,11,12,13]. Problematically, the primary outcomes in CSR populations depend on pain measurements, which are subjective in nature, and it is rare that investigations include measurements of neurophysiological outcomes to demonstrate improvement in nerve root function concomitant with pain improvements [8]. One exception to this is the trial by Moustafa and Diab [14] where they used three different cervical traction setups in an attempt to identify the optimum angle of combined distraction with flexion, neutral, or extension angles. The authors identified that distraction combined with slight head extension was found to be associated with the best improvement in neurophysiological measures in patients with cervical radiculopathy.

Regarding the development of signs and symptoms of CSR, the patho-anatomy of the vertebra and disc is not the only cause of a given patient’s pain; it is likely that the patho-anatomy, inflammatory mediators, functional disturbances, and altered spine alignment all interact to produce clinical symptoms [15]. In this regard, various studies point to the fact that biomechanical dysfunction of the spinal column, as seen with altered cervical sagittal plane alignment, results in degenerative changes in the muscles, ligaments, and bony structures [4,5,6,7]; altered spine alignment coupled with degenerative spine changes will increase the stress and strain on the neural elements potentially leading to and increasing the magnitude of neurologic dysfunctions in CSR [16,17,18,19,20,21]. Clinically, the goals of CSR patient care include sagittal plane alignment improvement in surgical [19,20,21,22] and conservative [13,14,15,16,17] settings. Regarding the conservative care setting, it is rare that investigations seek to address the radiographic alignment of the sagittal cervical spine as an outcome measure or predictive variable in CSR patients; [14,16] this may be due to the fact that the vast majority of conservative care techniques do not have the capability to significantly improve the shape and magnitude of the cervical lordotic curve [23,24,25]. The exception to this rule is three-point bending extension traction devices which are known to increase cervical curvature following a program of consistent care over the course of 8–12 weeks [25].

In an original collection of case studies, Pope [23] first incorporated a counter-stressing strap system (front pull pulling posterior-anterior in the posterior aspect of the cervical spine) to cervical extension traction with slight distraction on the skull, drawing attention to the possibility of cervical sagittal curve correction by a ‘so-called 2 way’ cervical extension traction. Later, in a non-randomized clinical trial, Harrison et al. [24] evaluated the effect of this three-point bending (two-way) cervical traction on restoring the sagittal curve in a chronic neck pain population without radiculopathy; they reported a significant increase in cervical lordosis and reduction in pain intensity.

In a pilot randomized trial looking at cervical spine disco-genic radiculopathy without spondylotic changes, Moustafa et al. [16], documented that cervical extension traction, using a novel cervical orthotic, improves the cervical lordosis and improves pain, disability, and neurophysiology. Regarding conservative care for CSM, cervical spine traction in flexion and distraction is one of the most commonly performed and investigated procedures, but this technique is not conducive to improving the abnormal cervical lordotic curve [2,8,9,10,11,12,13]. Furthermore, previous trials, [14,23,24,25] testing the effects of three-point bending types of cervical extension traction, have not clarified the relationship of cervical spine correction and its influence or effect on nerve root function and pain responses associated with improving an abnormal cervical lordosis in CSR patients. In terms of neurophysiological outcomes, dermatomal somatosensory evoked potentials (DSSEPs) can provide reliable information about segmental nerve root function and DSSEPs have been identified to correspond to clinical symptoms more closely than other electrophysiological examinations [26,27].

While it is known that the conservative management of cervical spondylotic radiculopathy is beneficial and multiple therapies (multi-modal) should be used simultaneously, a recent systematic literature review concluded that neck and arm pain improvements were ‘trivial’ at best and that further research into the best methods for specific patient populations is needed [8]. Accordingly, in properly selected patients, cervical curve restoration interventions might offer unknown benefits. Thus, the present randomized controlled trial was undertaken to investigate the neurophysiological and pain response outcomes of three-point bending (two-way) traction compared to standard care in patient cases with lower cervical spine CSR, chronic pain, and with a verified hypo-lordosis of the cervical spine. The primary hypothesis of this study was that cervical lordosis restoration will have short and long-term effects on DSSEPs and pain outcomes in CSR patients.

## 2. Materials and Methods

A prospective, investigator-blinded, parallel-group, randomized clinical trial was conducted at a research laboratory in our university and was retrospectively registered with ClinicalTrials.gov (NCT05547997) accessed on 20 September 2022. The reason for retrospective trial registration was that legislation in Egypt only required local registration for clinical trials at the time of study design and this is what was conducted initially by prospectively registration in a non-WHO-approved registry. Recruitment began after approval was obtained from the Ethics Committee of the Faculty of Physical Therapy, Cairo University with the ethical approval No Cairo23-987-12 M.S. All participants signed informed consent prior to data collection. 

### 2.1. Patients

Thirty patients with lower cervical spine CSR participated in this study. There were nineteen females and eleven males ranging from 40 to 50 years of age. We randomly assigned the participants into a study group and a comparative control group. The study group, receiving three-point bending cervical traction, included nine females and six males. The comparative treatment (control) group consisted of five males and 10 females. 

### 2.2. Inclusion and Exclusion Criteria

Patients were included if they had unilateral radiculopathy due to spondylotic changes of the lower cervical spine (C5–C6 and/or C6–C7). Participants were screened prior to inclusion by measuring their lateral cervical radiographs for a cervical absolute rotation angle (ARA) formed by two lines intersecting from the posterior body margins of C2–C7. If the ARA angle was less than 20°, then participants were included in the study and determined to have hypolordois of the cervical curve [28,29]. In addition to cervical lordosis ≥20°, exclusion criteria included: (1) central spinal canal stenosis; (2) rheumatoid arthritis; (3) vestibulobasilar insufficiency; (4) osteoporosis; (5) any disorder that might affect the DSSEPs such as thoracic outlet syndrome, carpal tunnel syndrome, cubital tunnel syndrome, etc.; (6) patients who had received surgical treatment for CSR or neck injury; (7) patients with cervical spinal instability; (8) patients with comorbid severe primary diseases such as cardiovascular disease, cerebrovascular disease, diseases of the liver, kidneys, or hematopoietic system; (9) patients who were suffering from any malignant disease as well as those unable to tolerate the cervical extension position with increased axial pain and/or radiculopathy.

The diagnostic criteria of CSR in the current study included: pain and numbness in the distribution of spinal nerve roots C6, and/or C7; additionally, the brachial plexus tension test or foraminal compression test had to be positive. In all participants, the location of symptoms (e.g., dermatomal pain or neurological deficit) matched the evaluated nerve root. Moreover, the clinical manifestations and imaging findings were consistent with their clinical syndromes. Both plain cervical spine radiographs and MRI were used to assist in CSR diagnosis and rule out other diseases, such as disc herniation, infection, and tumor. Lastly, participants had to have side-to-side amplitude differences of 50% or more in their DSSEPs measurement, a duration of symptoms of more than 3 months, and a “present” pain score of 4 or higher on a scale of from 0 to 10. Included participants were randomly assigned to an intervention group (*n* = 15) or control group (*n* = 15) using a random number generator and were restricted to permuted blocks of different sizes, with the researcher blinded to the sequence designated for each person.

### 2.3. Treatment Procedures

Both groups (study and control) were provided standard comparative care to improve pain intensity and reduce muscle tension that might be responsible for a reduction in cervical lordosis; this standard care included stretching exercises and infrared radiation (IR). Additionally, the study group was treated with three-point bending cervical extension traction. All participants received their respective interventions, in a controlled environment, for three days per week for ten weeks for a total of 30 sessions. Participants were followed for 3 months and 2 years at which times re-assessments were performed.

Cervical traction procedure: The study group received three-point bending cervical extension traction following the protocol of Harrison et al. [24]. The head halter was fixed posteriorly to cause slight distraction, retraction, and slight extension and at the same time a front anterior strap had weight applied over a pulley that allows transverse traction load to be applied to the apex of the participants’ cervical curve alteration. Following the findings of Moustafa et al. [14], the angle of the posterior head harness pull was positioned, relative to vertical, 5–30° backward in order to cause slight extension and distraction as this position was found to be associated with the best improvement in DSSEP’s in patients with radiculopathy. Weights started at 15 lbs. (6.8 kg) on the anterior strap and increased over consecutive visits to patient tolerance or a maximum of 35 lbs. (15.9 kg). The duration of each session started at approximately three minutes and increased to one minute per session until reaching the goal of 20 min per session. Figure 1 represents the cervical 2-way traction method.

Stretching exercises: Exercises were performed in the following order: (1) stretching towards lateral flexion for the upper part of the trapezius; (2) ipsilateral flexion and rotation for the scalene, and (3) flexion for the extensor muscles. Each maneuver was held for 30 s as this is an optimum time to not create an alteration in the evoked potentials [30]. Each stretch was repeated three times. Patients performed the stretching program three times a week for 10 weeks and this treatment took approximately 10 min per session [31].

### 2.4. Outcome Measures

A series of outcome measures were obtained at three intervals: (1) baseline; (2) one day following the completion of 30 visits after 10 weeks of treatment; (3) at the 3-month follow-up after the 10-weeks of treatment re-evaluation; (4) at two years follow-up after the 10-weeks of treatment re-evaluation (1-year and 9-months after the 3-month follow-up). The sequence of measurements was identical for all participants. Radiographic cervical sagittal alignment of lordosis (ARA C2–C7) and neurophysiological findings were the primary treatment outcomes, whereas, the numerical pain rating scale (NPRS) variable was the secondary measure.

### 2.5. DSSEPs

The main outcome measure used to assess the nerve root function was the peak-to-peak amplitude of dermatomal somatosensory evoked potentials (DSSEPs). An electromyogram device (Tonneis neuroscreen plus version 1.59, Erich Jaeger, Inc., Rheda-Wiedenbrück, Germany) was used to measure this variable for all patients before starting the treatment, at the end of 10 weeks, at a follow-up of three months, and the long-term follow-up period of two years. All testing procedures for DSSEPs were conducted following the protocol of Liguori et al. [32] The patient was lying supine on a softly padded table with a pillow under their head and knees. After the skin was abraded and cleaned with alcohol, the stimulating electrodes were placed overlying dermatomes of C6 (about 7 cm above the styloid process of the radius) and C7 between the second and the third metacarpal bones and at C8 (medial side of the hand). Figure 2 demonstrates this procedure. A bipolar electrode was used for stimulation with an inter-electrode distance of 2.5 cm with the stimulation cathode placed proximally. The sensory threshold for the electrical stimulation was determined by increasing the intensity of the electrical current until the patient reported its sensation, tolerable and painless stimulus intensity was set at 2.5 times above this level. The recording was made with 9 mm diameter tin/lead electrodes affixed with electrolyte paste to the abraded skin. The recording electrodes were placed at C3 and C4 (between C3 and P3 and C4 and P4 of the international EEG 10–20 system), while the reference electrode was placed at Fz and the ground electrode at Fpz. See Figure 3. The cortical responses were amplified, averaged and displayed using an analysis time of 50 ms and a filter setting of 2 Hz to 1 kHz was used in this study. After the stimulation was performed and traces were superimposed to ensure reproducibility, negative near-field potentials were detected to measure the peak-to-peak amplitude.

Cervical Lordosis: Cervical spine, standing, and lateral X-rays were obtained for each participant at four time periods: at baseline, following 10-weeks or 30 treatment sessions, at the 3-month follow-up, and at final follow-up of 2 years. The participants were asked to adopt a relaxed neutral posture and look straight forward as if staring into their own eyes in a mirror; this procedure has been investigated and has good to excellent examiner reliability [24]. The cervical lordosis was measured using the posterior body tangent method where a line is drawn along the posterior aspect of the C7 vertebral body and the angle of the curve is measured with an intersecting line drawn along the posterior vertebral body margin of C2; this is termed the absolute rotation angle or ARA of C2–C7. The ARA C2–C7 lordosis was measured using a standard protractor and sharp X-ray pencil; this measurement method has excellent examiner reliability [33].

Pain intensity: Neck and arm pain intensity were measured using the numerical pain rating scale (NPRS), which is considered a valid and reliable scale [34]. The patients were asked to place a mark along the line to denote their pain level; 0 reflecting ‘‘no pain’’ and 10 reflecting the ‘‘worst pain’’.

### 2.6. Sample Size Estimation

To determine the required number of participants needed in this study, estimates of mean and standard deviations (SD) were collected from a pilot study consisting of 10 participants who received the same program. The mean differences and SD of the ARA C2–C7 and peak-to-peak amplitude of DSSEPS for different levels C6, 7, and 8, were: ARA, −7 (SD 1.2); C6: –0.6 (SD 0.1); C7: –0.7 (SD 0.2); C8: –0.6 (SD 0.3), respectively. These values were used to calculate the sample size separately for each of the primary outcomes by applying a Bonferroni correction to adjust the significance level. The largest value of the sample size was then considered the final sample size for the trial. Accordingly, at least 14 participants in each group, given a statistical power of 80%, were needed in the current study. The sample size was enlarged by 10% to account for potential dropouts.

### 2.7. Data Analysis

Descriptive statistics were calculated including mean ± standard deviation (SD) for age, height and weight. For between-group repeated measures an analysis of covariance was used: Our model used the group as an independent variable, time as the repeated measurement, and group × time as the interactive variable. In order to assess between-group differences, participants’ baseline variable outcomes were used as covariates; where each participant’s value was subtracted from the population mean. The Bonferroni correction was used if we identified group × time interactions, (*p* < 0.05). In order to assess any possible linear correlation fits between variables, Pearson correlations between ARA C2–C7 and peak-to-peak amplitude values of DSSEPs, and ARA C2–C7 and pain scores were determined. The correlation findings were compiled into a pre-study set and a post-study set. The level of significance was set at *p* < 0.05.

## 3. Results

The study group, consisting of fifteen patients receiving the new extension traction, was compared with the fifteen control participants who received standard care only (IR and stretching exercises). Patient demographics are shown in Table 1 where it is shown that the two groups were statistically matched for age, weight and height. Patient retention throughout the study is shown in Figure 4.

### 3.1. Pain Outcomes

At 10 weeks of treatment, pain intensity was significantly improved (*p* < 0.0001) for both the study and control groups; indicating a reduction in pain due to interventions in both groups. Using Tukey’s Multiple Comparison Test, we identified that the study group’s pain was unchanged at the 3-month follow-up compared to the 10-week values; *p* > 0.05. While at the 2-year follow-up the study group’s pain continued to improve with a statistically significant decrease in pain at 2 years compared to 3 months, mean difference of 1.1 and *p* < 0.05. In contrast, the control group revealed a significant increase (worsening) in the mean pain at 3 months and 2 years compared to their 10 weeks of treatment evaluation; *p* < 0.05 at 3 months. The between-group analysis identified that the study group had statistically significant reductions in pain compared to the control group at each of the three follow-up measurements; *p* < 0.0001. See Table 2 and Figure 5.

### 3.2. ARA C2-C7 Cervical Lordosis

Regarding the cervical lordosis (ARA C2–C7), in the study group the one-way ANOVA (baseline versus 10 weeks), identified an increased cervical lordosis ARA C2–C7, *p* < 0.0001 and F = 49.8. In contrast, the control group was identified to have no statistical change in cervical lordosis; (*p* > 0.05). For the study group, using Tukey’s Multiple Comparison Test, the ARA C2–C7 was unchanged at 3 months and 2 years in comparison to the 10-week data (mean difference of 1.333 at 10-weeks; *p* > 0.05). In contrast, for the control group, the post-test was not calculated due to insignificant differences; *p* > 0.05. The between-group analysis identified that the study group had statistically significant increases in ARA C2–C7 cervical lordosis compared to the control group at each of the three follow-up measurements; *p* < 0.0001. See Table 2 and Figure 6.

### 3.3. DSSEPs

The repeated measures one-way ANOVA, comparing initial DSSEPs to 10-week treatment values, identified statistically significant improvements for both groups (*p* < 0.0001). A Tukey’s Multiple Comparison Test revealed significant increases in the mean of the post-test compared with pretreatment values for both the study and controls. However, only in the study group did the post-test reveal insignificant changes in DSSEPs at 3-month and 2-year follow-ups compared to the 10-week data; *p* > 0.05. In contrast, at 3-month and 2-year follow-ups the control groups DSSEP measurements regressed back to baseline values. The between-group analysis identified that the study group had statistically significant improvements in the DSSEPs for all three nerve root levels compared to the control group at each of the three follow-up measurements; *p* < 0.0001. See Table 3 and Figure 7 and Figure 8.

### 3.4. Correlations

All correlation results for ARA C2–C7 lordosis and pain and the DSSEPs at each of the three levels are presented as (1) a baseline correlation and (2) for follow-up treatment data at the 2-year mark. At baseline, for both groups, increased cervical lordosis was inversely correlated to pain intensity (r= −0.49; *p* = 0.005); however, this inverse correlation was only maintained at follow-up for the study group receiving traction (r = −0.6; r= *p* = 0.01). See Table 4. We identified a linear correlation between initial DSSEPs and ARA C2–C7 for both groups at each of the three nerve root levels C6–C8 (r = 0.65, r= 0.57, r= 0.8, *p* < 0.0001). Whereas this linear relationship between ARA C2–C7 became insignificant in the control group but was maintained in the study group at a 2-year follow-up at C6 (r = 0.55; *p* = 0.033). In contrast, both groups were found to have significant correlations at the 2-year mark for C7 and C8 nerve root DSSEPs (*p* < 0.001). See Table 4

### 3.5. Medication and Alternative Therapy Usage

At the 2-year follow-up, participants were asked if they were using alternative (non-surgical) therapies and/or medications to aid in managing the frequency and intensity of pains. Table 5 reports these interventional therapies utilized by the participants in the two groups (Study Group and Control Group) tracked at the 2-year follow-up. The data are reported by an individual participant in each group that the information was obtained from and not the number of people in each group using each intervention. Thus, 11 total participants were using medications and therapies (nine participants in the control group and two participants in the study group) indicating alternative services and medications were used by 4.5 times more participants in the control group and they were using a greater number of services. Table 5.

## 4. Discussion

This study compared outcomes of cervical spondylotic radiculopathy (CSR) in a group receiving three-point bending cervical extension traction combined with neck stretches and IR to a group receiving neck stretches and IR only. We had hypothesized that the study group receiving traction would show cervical curve correction resulting in short and long-term benefits on neurophysiological findings and improved pain. The differences between our study and control groups’ short and long-term radiographic, DSSEPs, and pain parameters indicate that this hypothesis is supported. This study provides objective evidence that sagittal cervical curve malalignment, and not just pathoanatomy, influences nerve root function and pain.

### 4.1. Cervical Lordosis Improvements

Concerning the cervical lordosis in the study group, a primary finding was a significant increase in the ARA C2–C7 (mean 7.5°) after 10 weeks of three-point bending traction treatment with no significant loss of lordosis at 3-month and 2-year follow-up. In contrast, the control group receiving IR and neck stretches revealed no significant differences in cervical lordosis between baseline, 10 weeks of treatment, 3-month, and 2-year follow-up measurements. Our study group’s results are in agreement with a previous non-randomized controlled trial on three-point bending traction conducted by Harrison et al. [24] Here, [24] three-point bending cervical traction combined with cervical manipulation was found to improve segmental and global cervical lordosis by a mean of 14° in thirty-seven sessions over the course of 8 to 10 weeks. In a pilot randomized trial on cervical radiculopathy due to disc herniation, Moustafa and colleagues [16] demonstrated that their group receiving a novel extension traction device termed the Denneroll, was found to have an improvement in lordosis of approximately 13° after 10 weeks of care. An explanation for the reduced cervical curve improvements (about 50% less) found in the current study compared to the Moustafa et al. [16] and Harrison et al. [24] investigations is likely a result of the different types of spine disorder populations being studied; chronic neck pain vs. CSR patients in the current study and the modification to the extension traction position for CSR patients. Though our trial is the first to assess lordotic improvements in a specific population with CSR receiving three-point bending cervical extension traction, the results are qualitatively comparable to previous investigations reporting cervical curve correction with these types of traction [25].

Loss of cervical lordosis is often attributed to muscles spasm. Thus, it may be speculated that our study group’s increased lordosis was attributed to the relief of muscle spasms and or tightness. However, we found no statistically significant differences in the control group’s cervical lordosis who were subjected to neck stretches and IR; which should also reduce muscle spasm/tightness. The lack of a cause-and-effect association between muscle spasm and hypo-lordosis in our study is consistent with a study of acute and chronic neck pain patients by Helliwell et al. [35] and with the biomechanical investigation performed by Fedorchuk et al. [36].

### 4.2. Pain Improvements

Our study findings offer initial encouragement for pain management in CSR patients using conservative care. For our control group, the transient short-term effect of traditional exercises and IR alone are in agreement with Ylinen et al. [1] who conducted a study to compare the effects of manual therapy and stretching exercise on neck pain and disability. The difference in effectiveness between the two treatments was minor and low-cost stretching exercises were recommended in the first instance as an appropriate intervention to relieve pain, at least in the short term. The randomized trial by Levoska and Keinänen-Kiukaanniemi [37] also found that stretching, light exercises, clay, and massage treatments reduced the occurrence of chronic neck pain. Regarding the efficacy of traction therapy on the outcomes of CSR, a recent randomized trial with a 3-month follows up found that distraction traction therapy provided improvements that reached minimally important clinical differences in about 50% of treated patients [38]. However, in a systematic literature review, Colombo and colleagues [13] identified that, compared to matched controls, the reduction in pain intensity after traction was statistically significant but did not reach meaningful clinically important differences at follow-up. The conflicting, transient, and limited effect of conservative therapies for CSR management reported in some trials is likely multi-factorial and may be attributed to the unique variables of the individual patient. For example, sustained postural imbalance, represented by cervical hypolordosis or kyphosis, causes increased and altered mechanical loading [4,5,6,7,8,20,21,22]. Once abnormal sagittal cervical alignment becomes established and maintained beyond a critical threshold, the result will be an increase in the probability of pathologies in both the soft and hard tissues of the spine [4,5,6,7,20,21,22]. To this point, in both our study and control groups, we identified a statistically significant negative correlation between cervical lordosis and neck pain for the pre-treatment data (r = −0.49). In other words, as the cervical lordosis became straighter, the pain intensity increased. 

Of importance, comparing the 10-week to the 3-month data, there was a correlation between the amount of change in lordosis and pain intensity for the traction group; while there was an insignificant association for the control group. These findings indicate that the improvement in pain intensity in the study group at 3-month and 2-year follow-up is probably a result of restoring the cervical lordosis. Overall, our findings support a mechanical relationship between loss of lordosis and pain intensity in this CSR population, particularly at long term follow-up. This mechanical relationship between loss of cervical lordosis and neck pain has previously been identified in two separate investigations. Both McAviney et al. [28] and Harrison et al. [29] identified moderate to good sensitivity and specificity for a hypo cervical lordosis (less than 20°) to discriminate between normal controls and chronic neck pain subjects without significant spinal degeneration. In contrast, in a prospective study of 107 volunteers aged over 45 years with moderate-severe degeneration, Grob et al. [39] examined the correlation between the presence of neck pain and alterations in cervical lordosis concluding that the presence of such structural abnormalities in the patient with neck pain is not related to their cause of pain.

The discrepancy and conflict regarding cervical lordosis found in the results obtained by the previous authors [28,29,39] cannot be directly compared with our current study for several reasons. First, the previous studies [28,29,39] were cross-sectional correlation studies without the ability to ascribe cause and effect. Second, the selection criteria for patient inclusion in the previous studies were patients complaining of primary lower extremity pain in the Grob et al. [39] study and acute and chronic neck pain patients without CSR in the McAviney et al. [28] and Harrison et al. [29] study. In the current study, after 30 sessions, the study group’s cervical curve closely approximated the 20° benchmark as reported previously [28,29]. However, note-worthy is that over the 2-year time period, the study groups’ ARA-lordosis is becoming slightly decreased compared to the 10-week post-treatment value. It is interesting to speculate the need for further corrective interventional care in this group to maintain the ARA above the 20° mark; future studies are needed to evaluate multiple 10-week programs of care and supportive care over the course of 2-year follow-up in an effort to maintain the cervical curve above the 20° mark.

### 4.3. DSSEPs Improvement

We used DSSEPs to measure depressed and improved nerve root function resulting from CSR. DSSEPs overcome the inherent problems associated with mixed nerve stimulation as in the case of F wave measures and mixed nerve SSEPS will be minimized. At 10 weeks of treatment, we found statistically significant improvements in DSSEPs for both groups (one-way ANOVA, *p* < 0.0001). However, at 3-month and 2-year follow-ups, the control group’s values regressed back to baseline values whereas the traction group continued to show statistically significant improvements. Our findings indicate only a transient effect on DSSEPs for stretching exercises and infrared radiation when used alone for the treatment of CSR populations visible at the 10-week immediate post-treatment follow-up. Qualitatively, our findings are in agreement with the clinical trial on CSR by Moustafa and Diab [14] where they used three different cervical traction setups in an attempt to identify the optimum angle of combined distraction traction. The authors identified that distraction combined with slight head extension was found to be associated with the best improvement in neurophysiological measures in patients with cervical radiculopathy and this result was maintained at 1-year follow-up. Though these authors discuss their findings relative to an abnormal cervical lordosis and the extension traction position likely benefits the lordotic configurator; no radiographic data was supplied [14].

Significantly, we identified a linear correlation between initial DSSEPs and cervical lordosis (ARA C2–C7) for both groups at initial evaluation (r = 0.65; *p* < 0.0001); whereas, this relationship was only maintained in the study group at the final follow-up for all measured cervical root levels. Thus, our findings support a relationship between abnormal cervical lordosis and altered neurophysiological deficits on the one hand and that the consequent improvement in neurophysiology is related to the restoration of cervical lordosis. Still, it seems logical and, is generally accepted, that ventro-flexion traction (especially for the lower cervical spine) is more beneficial in improving the nerve root function in CSR due to its effects on the intervertebral foramen [12,13,38]. For example, Wainner and Gill [40] evaluated the nonsurgical treatment of cervical disc herniations with flexion distraction and reported that flexion distraction might be an effective therapy in the treatment of cervical disc herniation and improving neural function as indicated by a reduction in pain. Though contradictory as it seems, our findings support a strong correlation between lordosis increases and peak-to-peak amplitude of DSSEPs for pre-and-post manipulating data. To the best of our knowledge, this is the first study to explicitly examine these relationships in detail in a clinical trial on CSR patients. 

Mechanically, the current study findings make sense and agree with Schnebel et al. [41] who investigated the role of spinal flexion and extension in changing nerve root compression (transverse load). It was found that the amount of compressive force and tension in the nerve root was increased with flexion of the spine and decreased with the extension of the spine. This tension and compression may adversely affect the CNS and nerve root function due to the absence of any perineurium, the primary load-carrying structure [17,18]. The observations of Abdulwahab and Sabbahi [42] also correlate well with this mechanical explanation. These authors [42] found that neck retraction appeared to increase the H reflex amplitude in patients with radiculopathy; the opposite effect was found with cervical flexion posture.

The conflicts found in the results of the previous investigations and the current study findings regarding nerve root function and flexion distraction vs. two-way extension traction can be explained in two ways. Previous studies have referred to an increase in the volume of the intervertebral foramen as a direct cause of decompression; while simultaneously disregarding the adverse mechanical tension and shear experienced by the spinal cord and nerve roots as they make contact with any infringing pathology [17,18]. This concept is in agreement with Albert et al. [43] and supported by Brian et al. [44] who reported that although foraminal height and foraminal area increase significantly after anterior cervical discectomy and fusion in a patient with cervical radiculopathy, no correlation was found with relief of clinical symptoms. The second reason explaining the above conflict is that many studies [5] refer to the improvement in patient pain as a direct measure for improvement in the nerve root function; ignoring the fact that neurophysiological deficits in CSR often occur without overt pain or symptomatology. Pain seems to have a strong correlation only when there is inflammation, especially when the dorsal root ganglia are involved [45].

### 4.4. Limitations

The current study has several limitations, each of which points toward directions of future study. The primary limitations were the lack of investigator blinding and the sample was a convenient sample of patients with CSR rather than a random sample of the whole population. Further, the sample size was just above the minimum number for statistical significance with only 15 participants per group; 14 were needed. Larger sample sizes in future RCTs need to be performed to confirm or refute our findings; specifically, the 2-year follow-up in the control group where the sample size of 12 participants’ data was just under the minimum of 14 needed for robust statistical claims to be made. Ideally, it would have been beneficial to provide a 5-year follow-up of our population to truly understand the impact of cervical curve restoration in the long term. However, due to the smaller sample size of our trial (15 participants in each group), it was not possible to follow our patients past 2-years as we would not have had enough data for statistical analysis. Additionally, in terms of the existing conservative care literature on CSR outcomes in RCTs, it is clear that studies use 1–2 year follow-ups as their definition of ‘long term’. In fact, most CSR RCTs only offer 3-month to 1-year follow-up and it is rare that studies go on for 2 years and longer [2,8,9,10,11,12,13]. Still, future investigations should provide results at 1, 2, 5-year, and 10-year follow-ups using the type of extension two-way traction as reported in our investigation to truly understand the long-term results of curve correction in CSR. Lastly, biomechanical investigations via computer simulation would be beneficial in future experimental designs to understand the soft tissue deformation and strain/strain effects of three-point-bending extension traction methods for cervical curve restoration in patients with cervical spondylotic radiculopathy.

## 5. Conclusions

Our investigation identified that the correction of the cervical lordosis, in hypolordotic spines of patients suffering from CSR, had improved pain and neurophysiology. The group receiving three-point bending cervical traction attained a significant increase in cervical lordosis, improvement in their pain intensity, and nerve root function measured with DSSEPs. Follow-up measurement revealed stable improvement in all measured variables. These observed effects of sagittal curve correction offer insights to clinicians working with patients with cervical spine disorders such as chronic CSR. Future trials should continue to investigate the rehabilitation of the abnormal cervical curve in CSR populations focusing on larger sample sizes, who are the optimum candidates, what an adequate curve correction is, and longer follow-up time periods.

## Figures and Tables

**Figure 1 jcm-11-06515-f001:**
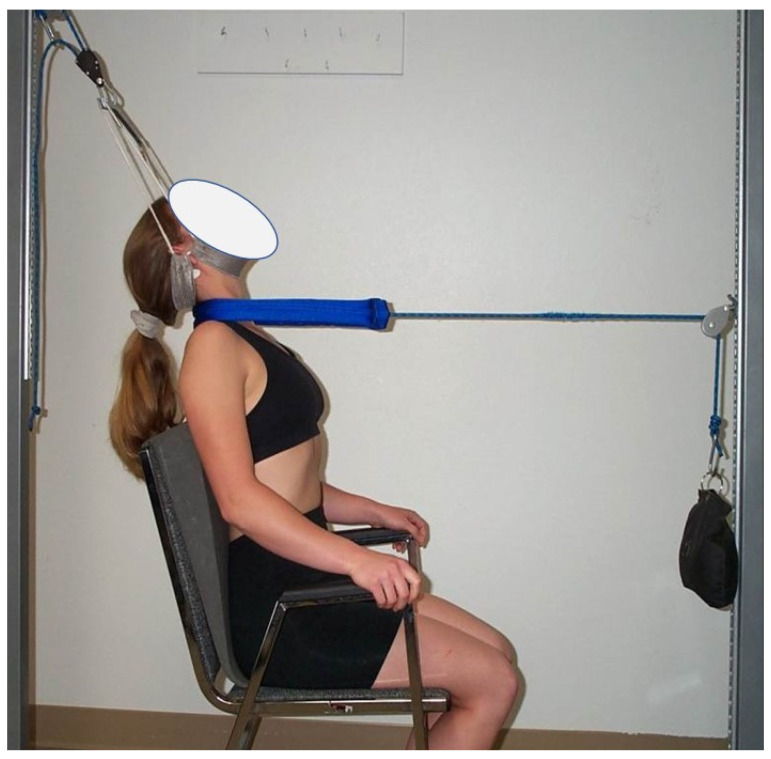
Three-point bending cervical traction. Photo reprinted with permission.

**Figure 2 jcm-11-06515-f002:**
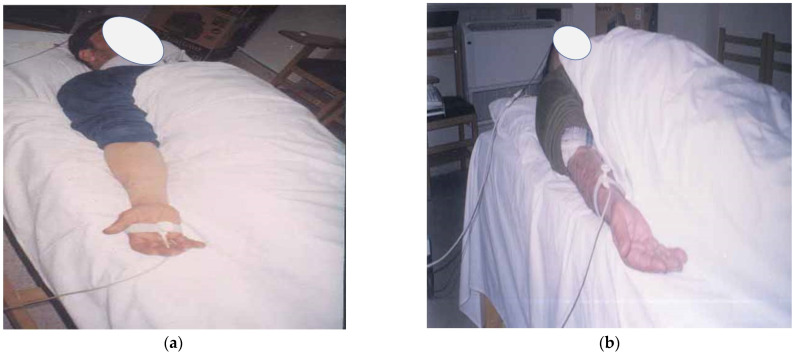
Location of stimulation sites indicated by arrow (**a**) for the C7 dermatome and (**b**) for the C6 dermatome.

**Figure 3 jcm-11-06515-f003:**
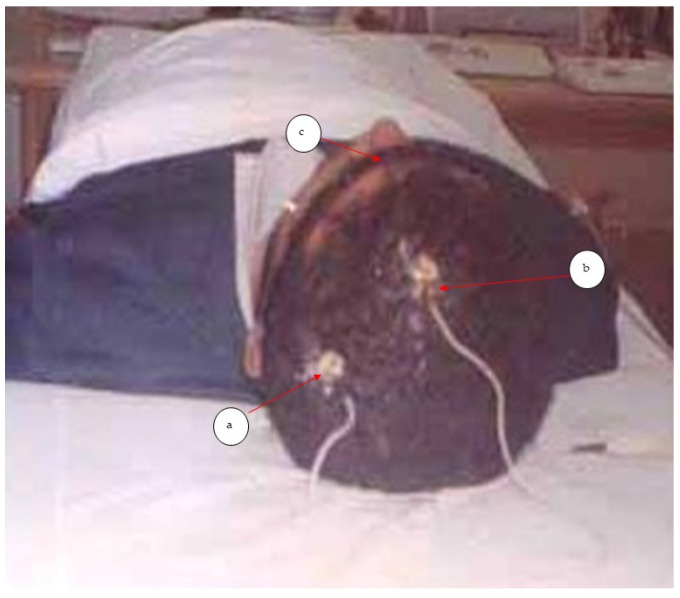
Sites of Recording: (**a**) active recording electrode at c3’, (**b**) reference electrode at Fz, and (**c**) grounding electrode at Fbz.

**Figure 4 jcm-11-06515-f004:**
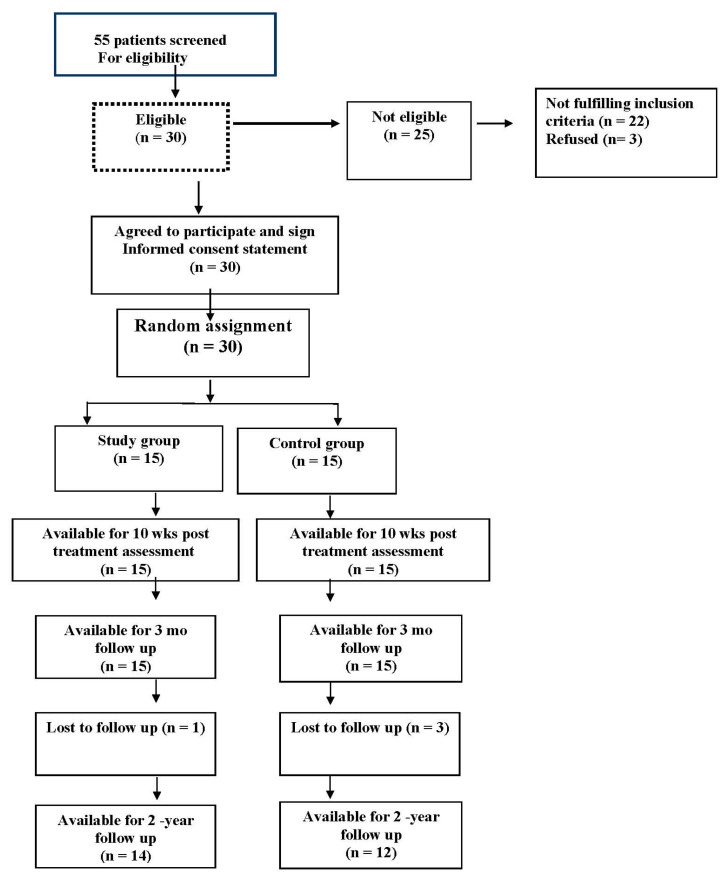
A diagram of patients’ retention and randomization throughout the study is shown.

**Figure 5 jcm-11-06515-f005:**
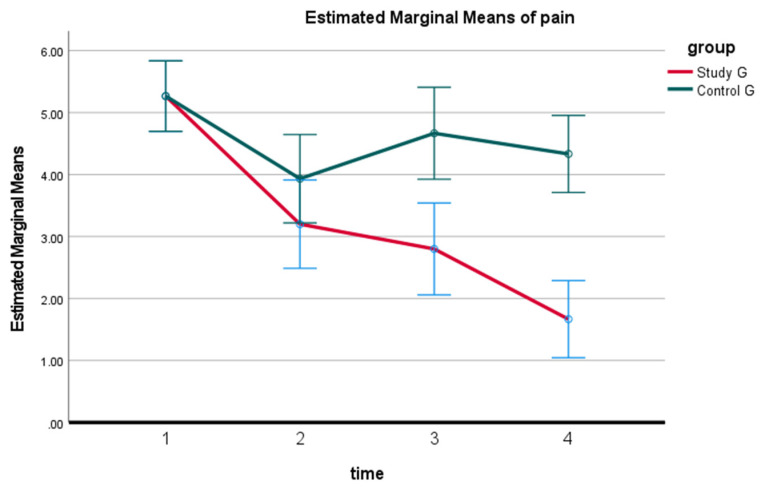
Differences in pain on the numerical rating scale (NRPS) during the study reported as mean ± SD for study and control groups at four time periods: baseline or pretreatment, after completion of the 10-week program, the 3-month follow-up, and the 2-year follow-up data. 1: pretreatment; 2: 10 weeks post-treatment; 3: at 3 months; 4: at 2-year follow-up.

**Figure 6 jcm-11-06515-f006:**
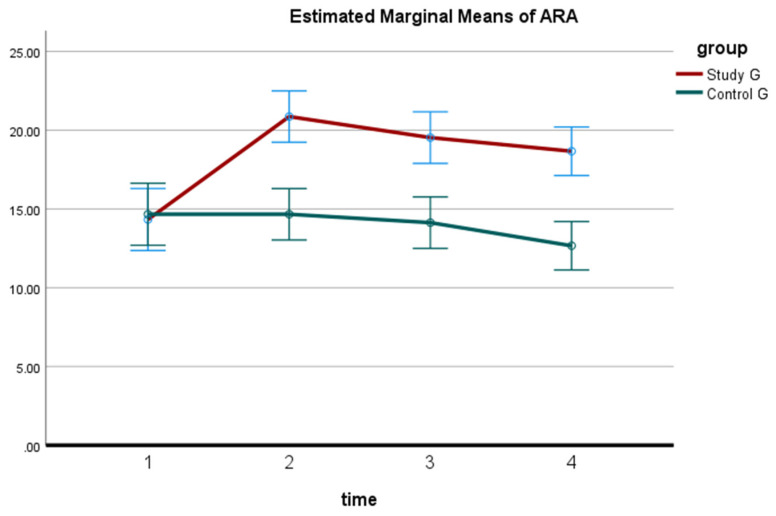
The cervical lordosis absolute rotation angle C2–C7 (ARA) for the control group and study group is shown as the mean ± SD. Four different time periods are shown: baseline or pretreatment, after completion of the 10-week program, the 3-month follow-up, and the 2-year follow-up. 1: pretreatment; 2: 10 weeks post-treatment; 3: at 3 months; 4: at 2-year follow-up.

**Figure 7 jcm-11-06515-f007:**
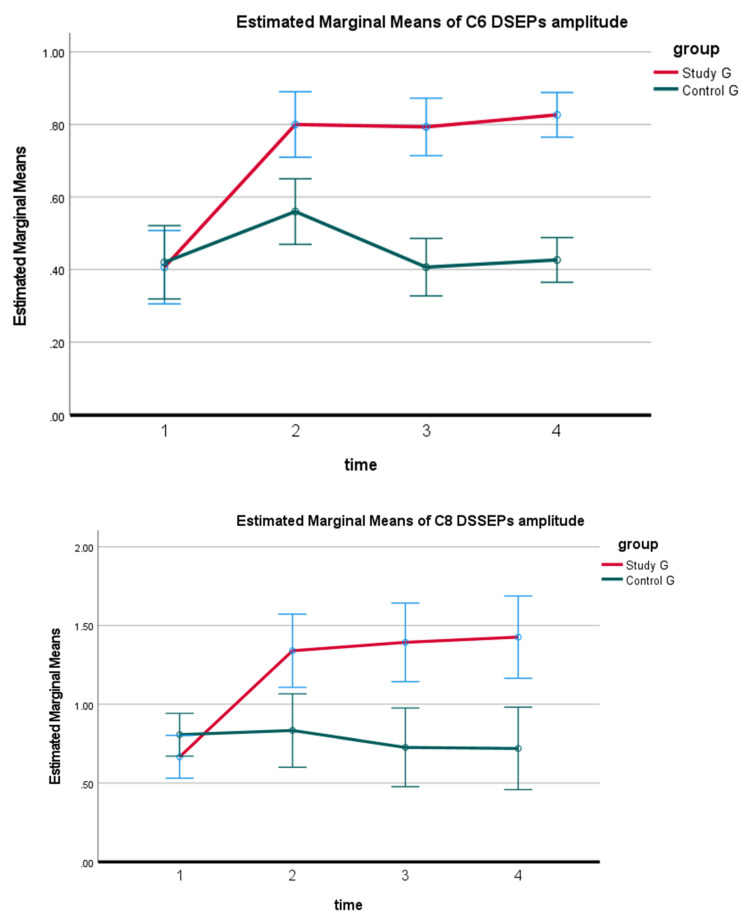
Mean and ± SD of the DSSEPS for study and control groups at four time periods: baseline or pretreatment, after completion of the 10-week program, the 3-month follow-up, and the 2-year follow-up.

**Figure 8 jcm-11-06515-f008:**
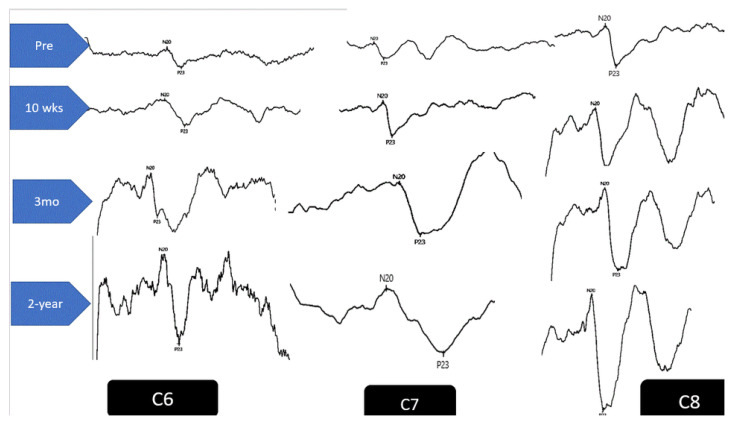
Example of DSSEPs at each of the levels C6–C8 at the four intervals of measurement for the study group (2-way traction).

**Table 1 jcm-11-06515-t001:** Baseline participant demographics.

	Study Group (*n* = 15)	Control Group (*n* = 15)	*p* ‡
Age (years)	46.3 ± 2.05	45.9 ± 2.1	0.5
Weight (kg)	73.3 ± 8.9	77.5 ± 9	0.2
Height (cm)	171.6 ± 5	168.3 ± 7.9	0.18
Male	6	5	0.7
Female	9	10	
Smoker	5	4	0.69
Non smoker	10	11	

‡ Two-sided two-sample *t*-test; SD: standard deviation, the values are mean (± SD) for age, height, weight and as the number for the term ‘other’.

**Table 2 jcm-11-06515-t002:** The results for the repeated measures one-way analysis of variance (ANOVA) for the absolute rotation angle (ARA) cervical lordosis and pain intensity in both groups.

Measures		Pretreatment	At 10 Weeks	At 3 Months	At 2 Years	*p*			Post Hoc Test (MD)	
						G	T	G × T		
ARA	S	14.3 ± 4.1	20.87 ± 3	19.5 ± 3.2	18.8 ± 2.1	<0.001	<0.001	<0.001	1 vs. 2	−3.26 *
	C	14.6 ± 3.5	14.7± 3.3	14.1 ± 3.1	12.3 ± 2.7				1 vs. 3	−2.33 *
		0.8 [−3.1–2.5]	<0.001 [3.8–8.5]	<0.001 [3.02–7.7]	<0.001 [3.7–8.2]				1 vs. 4	−1.16 *
Pain	S	5.26 ± 0.96	3.2 ± 1.26	2.8 ± 1.27	1.71 ± 1.2	<0.001	<0.001	<0.001	1 vs. 2	1.7 *
	C	5.47 ± 1.18	3.9 ± 1.43	4.6 ± 1.49	4.3 ± 1.2				1 vs. 3	1.5 *
		0.7 [−0.8–0.9]	<0.001 [−1.7–0.29]	<0.001 [−2.9–−0.79]	<0.001 [−3.5–−1.8]				1 vs. 4	2.3 *

Study group: SG; Control group: CG; * Statistically significant difference: *p*-value; MD: mean difference.

**Table 3 jcm-11-06515-t003:** The results for the repeated measures one-way analysis of variance (ANOVA) for the DSSEPs amplitudes in the study and control groups for three nerve root levels: C6, C7, and C8.

Measures		Pretreatment	At 10 Weeks	At 3 Months	At 2 Years	*p*			Post Hoc Test (MD)	
						G	T	G × T		
C6	S	0.41 ± 0.1	0.80 ± 0.19	0.79 ± 0.11	0.82 ± 0.14	<0.001	<0.001	<0.001	1 vs. 2	−0.267 *
	C	0.42 ± 0.2	0.56 ± 0.15	0.40 ± 0.15	0.42 ± 0.17				1 vs. 3	−0.18 *
		0.8 [−0.16–0.13]	<0.001 [0.11–0.37]	<0.001 [0.272–0.5]	<0.001 [0.31–0.48]				1 vs. 4	−0.21 *
C7	S	0.4 ± 0.1	1.18 ± 0.33	1.0 ± 0.37	1.1 ± 0.37	<0.001	<0.001	<0.001	1 vs. 2	−0.38 *
	C	0.69 ± 0.2	0.7 ± 0.18	0.52 ± 0.21	0.51 ± 0.18				1 vs. 3	−0.24 *
		0.05 [−0.26–0.02]	<0.001 [0.32–0.72]	<0.001 [0.30–0.76]	<0.001 [0.36–0.81]				1 vs. 4	−0.26 *
C8	S	0.6 ± 0.2	1.3 ± 0.5	1.4 ± 0.6	1.5 ± 0.6	<0.001	<0.001	<0.001	1 vs. 2	−0.35 *
	C	0.8 ± 0.2	0.9 ± 0.3	0.7 ± 0.3	0.7 ± 0.2				1 vs. 3	−0.32 *
		0.15 [−0.33–0.05]	0.005 [0.17–0.84]	<0.001 [0.30–1.1]	<0.001 [0.32–1.08]				1 vs. 4	−0.33 *

Study group: SG; Control group: CG; * Statistically significant difference: *p*-value; MD: mean difference; 1: pretreatment; 2: 10 weeks post-treatment; 3: at 3 months; 4: at 2-year follow-up.

**Table 4 jcm-11-06515-t004:** Pearson correlation between ARA C2–C7 and DSSEPS and between ARA and pain. Post-manipulating (post-treatment) data are shown for the 2-year follow-up compared to initial baseline data.

	Number of XY Pairs	r	*p*
ARA & DSSEP (C6) (baseline data)	30	0.65	<0.001 *
ARA & DSSEP (C7) (baseline data)	30	0.57	<0.001 *
ARA & DSSEP (C8) (baseline data)	30	0.8	<0.001 *
Post-manipulating data (C6) study control	15	0.55	0.033 *
	15	0.19	0.49
Post-manipulating data (C7) study control	15	0.74	<0.001 *
	15	0.62	<0.001 *
Post-manipulating data (C8) study control	15	0.8	<0.001 *
	15	0.58	<0.001 *
ARA C2–C7 and pain Baseline data	30	−0.49	0.005 *
Post-treatment data Study group	15	−0.6	0.01 *
Control group	15	−0.17	0.05 *

*p*: probability value; r: Pearson’s correlation coefficient; *: statistically significant difference.

**Table 5 jcm-11-06515-t005:** Medication and interventional therapies utilized by of the participants in the two groups (Study Group and Control Group) tracked at the 2-year follow-up.

Medication Utilization & Therapy Used
Control Group
• NSAIDs
• Tricyclic antidepressants
• NSAIDs, hydrotherapy
• NSAIDS, Acupuncture
• Tricyclic antidepressants, semi-hard cervical collar
• Tricyclic antidepressants, semi-hard cervical collar
• Tricyclic antidepressants, soft tissue massage
• Opioid medications
• NSAIDS, Ultrasound therapy
Study Group
• NSAIDs
• Cervical spine epidural steroids

The data are reported by an individual participant in each group in each individual row that the information was obtained from and not the number of people in each group using each intervention. Thus, 11 total participants were using medications and therapies (nine participants in the control group and two participants in the study group) indicating alternative services and medications were used by 4.5 times more participants in the control group and they were using a greater number of services.

## Data Availability

The datasets analyzed in the current study are available from the corresponding author on reasonable request.

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
