# Peer review of "The Efficacy of Cervical Lordosis Rehabilitation for Nerve Root Function and Pain in Cervical Spondylotic Radiculopathy: A Randomized Trial with 2-Year Follow-Up"

_jcm, 2022, doi:10.3390/jcm11216515_

Round 1

Reviewer 1 Report

I liked the manuscript. I think its important to describe the best methods for rehabilitiation so that clinitians can provide the best theurapies.

Its encouraging that the describet approach leaves permenent benifits in follow up investigations.

Perhaps a little bit more about the condition and why/how these participants developed condition would be nice, are there any known predispositions?

Reviewer 2 Report

1.      Please include all of the author’s emails after affiliation with name initials, except for the corresponding author based on MDPI format.

2.      At the end of your abstract, please provide a "take-home" message.

3.      Put the keywords in a new order based on alphabetical order.

4.      Please use lowercase font for all keywords based on MDPI format.

5.      It is unclear whether the author's something new in this work. According to evaluation, several published studies by other researchers in the past adequately explain the issues you made in the present paper. Please be careful to highlight in the introduction section anything really innovative in this work.

6.      Previous studies must be explained in the introductory part, including their work, innovation, and limits, to demonstrate the research gaps that will be filled in the current study.

7.      Why only 2 years follow up? It is not enough to evaluate the rehabilitation results. The findings not mature enough. Is there any explanation for only 2 years follow up? One main reason for rejecting the article.

8.      Biomechanical investigation via computer simulation is beneficial in clinical study for supporting the results. Would the authors provide it in the present study? It is very encouraged to improve the article. Also, further discussion related to computer study in healthcare improvement is need to discussed at least one sentence. It is important point that needs to be included by authors in the introduction and/or discussion section. Also, to support this explanation, the suggested reverence should be adopted as follows: Ammarullah, M. I.; Santoso, G.; Sugiharto, S.; Supriyono, T.; Kurdi, O.; Tauviqirrahman, M.; Winarni, T. I.; Jamari, J. Tresca Stress Study of CoCrMo-on-CoCrMo Bearings Based on Body Mass Index Using 2D Computational Model. Jurnal Tribologi 2022, 33, 31–8.

9.      To help the reader grasp the study's workflow more easily, the authors could include more visuals to the materials and methods section in the form of figures rather than sticking with the text that now predominates.

10.   What is the baseline of patient selection? Is there any protocol, standard, or basis that has been followed? It is unclear since the patient is very heterogeneous with a small number. The resonance involved impacts the present result makes this study flaws. One major reason for rejecting this paper.

11.   A comparative assessment with similar previous research is required.

12.   Mention further research in the conclusion section.

13.   The reference should be enriched with literature from the last five years. Literature published by MDPI is strongly recommended.

14.   In the whole of the manuscript, the authors sometimes made a paragraph only consisting of one or two sentences that made the explanation not clearly understood. The authors need to extend their explanation to become a more comprehensive paragraph. In one paragraph, it is recommended to consist of at least 3 sentences with 1 sentence as the main sentence and the other sentences as supporting sentences.

15.   Due to grammatical and language issues, the authors need to proofread the present work. This problem would use MDPI English editing service.

16.   Please ensure that the authors followed the MDPI format correctly; modify the current form and recheck, as well as any other problems that have been highlighted.

Reviewer 3 Report

Article:

The Efficacy of Cervical Lordosis Rehabilitation for Nerve Root Function and Pain in Cervical Spondylotic Radiculopathy: A Randomized Trial with 2-year Follow-up

The aim of these study was to analyze the effect of cervical lordosis rehabilitation with performing a “three-point bending cervical traction” technique. The authors’ included 30 patients in this prospective study. These patients, suffering from cervical spondylotic radiculopathy were divided in two groups, 14 patients were treated with the “three-point bending cervical traction” technique. 

The including and excluding criteria were well defined. The study design well demonstrated.

The authors conclude that the patient wit cervical traction treatment showed an important pain improved. In my opinion the main critic on this study is the small patients’ cohort of 30 patients. Nevertheless, the authors were able to show a non-operative treatment option in patients suffering from cervical spondylotic radiculopathy.

Round 2

Reviewer 2 Report

Reviewers greatly appreciate the efforts that have been made by the author to improve the quality of their articles after peer review. I reread the author's manuscript and further reviewed the changes made along with the responses from previous reviewers' comments. Unfortunately, the authors failed to make some of the substantial improvements they should have made making this article not of decent quality with biased, not cutting-edge updates on the research topic outlined. In addition, the author also failed to address the previous reviewer's comments, especially on comments number 5, 7, and 8. Thank you very much for the opportunity to read the author's current work.
